# SPADE: Semi-supervised Anomaly Detection under Distribution Mismatch

**Jinsung Yoon, Kihyuk Sohn, Chun-Liang Li, Sercan Ö. Arik, Tomas Pfister**
*{jinsungyoon, kihyuks, chunliang, soarik, tpfister}@google.com*
*Google Cloud AI*

**Reviewed on OpenReview:** *https://openreview.net/forum?id=JwDpZSv3yz*

## Abstract

Semi-supervised anomaly detection is a common problem, as often the datasets containing anomalies are partially labeled. We propose a canonical framework: Semi-supervised Pseudo-labeler Anomaly Detection with Ensembling (SPADE) that isn't limited by the assumption that labeled and unlabeled data come from the same distribution. Indeed, the assumption is often violated in many applications – for example, the labeled data may contain only anomalies unlike unlabeled data, or unlabeled data may contain different types of anomalies, or labeled data may contain only 'easy-to-label' samples. SPADE utilizes an ensemble of one class classifiers as the pseudo-labeler to improve the robustness of pseudo-labeling with distribution mismatch. Partial matching is proposed to automatically select the critical hyper-parameters for pseudo-labeling without validation data, which is crucial with limited labeled data. SPADE shows state-of-the-art semi-supervised anomaly detection performance across a wide range of scenarios with distribution mismatch in both tabular and image domains. In some common real-world settings such as model facing new types of unlabeled anomalies, SPADE outperforms the state-of-the-art alternatives by 5% AUC in average.

## 1 Introduction

Anomaly detection has numerous real-world applications, including identification of manufacturing defects, network security threats, and financial fraud (Chalapathy & Chawla, 2019; Ahmed et al., 2016; Vanerio & Casas, 2017). Anomaly detection can be considered in different settings. One is the fully-supervised setting, where the labels for all samples are available, for both normal and anomalous samples (Chawla et al., 2002; Estabrooks et al., 2004; Hwang et al., 2011; Barua et al., 2012). This setting is typically addressed with specialized approaches for data imbalance, e.g. weighted loss functions or resampling methods. An important special case of this fully-supervised setting is when only labeled normal samples exist (Schölkopf et al., 1999; Tax & Duin, 2004; Ruff et al., 2018; Golan & El-Yaniv, 2018; Sohn et al., 2021; Li et al., 2021), for which one class classifiers (OCCs) (e.g. with SVM (Schölkopf et al., 1999) or auto-encoder (Ruff et al., 2018)) and Isolation Forest (Liu et al., 2008) are popular approaches. Despite being widely-studied, the challenge towards the real-world use for these supervised settings is their tedious labeling requirement. At the other extreme, there is the fully unsupervised anomaly detection setting where no labeled data is available (Breunig et al., 2000; Liu et al., 2008; Zong et al., 2018; Bergman & Hoshen, 2019; Yoon et al., 2022). While the labeling costs can be entirely eliminated for this setting, the performance degradation is often significant compared to the supervised setting (Bergman & Hoshen, 2019; Zong et al., 2018), limiting its applicability for deployment.

To achieve the best of both worlds, we focus on the semi-supervised anomaly detection setting, aiming to achieve high performance with a limited labeling budget. In previous works on semi-supervised anomaly detection (Zhang & Zuo, 2008; Bekker & Davis, 2020; Blanchard et al., 2010; Akcay et al., 2018; Görnitz et al., 2013; Ruff et al., 2020), some focus on the positive-unlabeled setting (Zhang & Zuo, 2008; Bekker & Davis, 2020), and others utilize one-class classifiers or adversarial training on semi-supervised learning (Görnitz et al., 2013; Akcay et al., 2018). Ruff et al. (2020) treats all unlabeled data as normal samples to

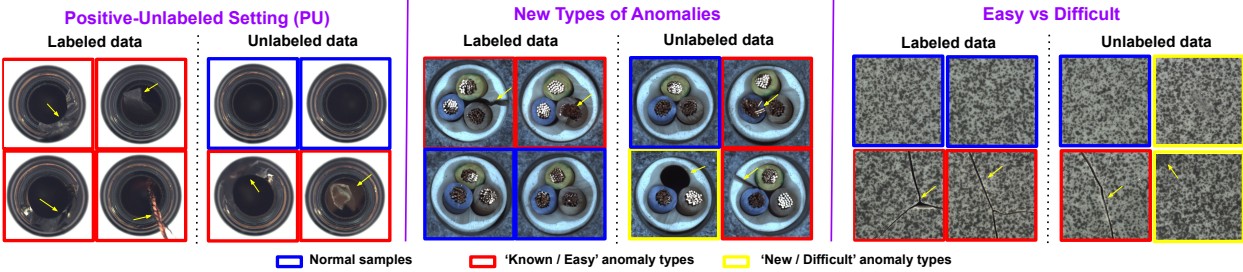

Figure 1: Three common real-world settings with labeled and unlabeled data coming from different distributions. (Left) Labeled data only include anomalous samples while unlabeled data have both anomalous and normal. (Middle) The anomaly is a new type (yellow boxes) which isn't in labeled data. (Right) Labeled data only have 'easy-to-label' samples while unlabeled data include 'hard-to-label' samples (yellow boxes).

construct an anomaly detector in semi-supervised settings. In addition, any semi-supervised learning method (even when they aren't developed for anomaly detection) can be adapted to the semi-supervised anomaly detection setting (Sohn et al., 2020; Chen et al., 2020a; Grill et al., 2020).

Most semi-supervised learning methods assume that the labeled and unlabeled data come from the same distributions (Sohn et al., 2020; Chen et al., 2020a; Grill et al., 2020). In other words, the subsets of the data are labeled such that sampling from the unlabeled data is randomly uniform. However, in practice, this assumption often does not hold: *distribution mismatch* commonly occur, with labeled and unlabeled data coming from different distributions. Some works (Kim et al., 2020) tackle this in a limited setting where only the label distributions are different (e.g., the anomalous ratio is 10% for training but 50% for testing), however, there are other more general real-world scenarios, as exemplified in Fig. 1. First, positive and unlabeled (PU) or negative and unlabeled (NU) settings are common, where the distributions between labeled (either positive or negative) and unlabeled (both positive and negative) samples are different (see Fig. 1(Left)) (Zhang & Zuo, 2008; Bekker & Davis, 2020). Second, additional unlabeled data can be gathered after labeling, causing distribution shift. For example, manufacturing processes may keep evolving and thus, the corresponding defects can change and the defect types at labeling differ from the defect types in unlabeled data (see Fig. 1(Middle)). In addition, for financial fraud detection and anti-money laundering applications, new anomalies can appear after the data labeling process, as the criminals adapt themselves. Lastly, human labelers are more confident on easy samples; thus, easy samples are more likely to be included in the labeled data and difficult samples are more likely to be included in the unlabeled data (see Fig. 1(Right)). For example, with some crowd-sourcing-based labeling tools, only the samples with some consensus on the labels (as a measure of confidence) are included in the labeled set.

As we experimentally demonstrate (in Sec. 5), standard semi-supervised learning methods (Sohn et al., 2020; Chen et al., 2020a; Grill et al., 2020) are sub-optimal for anomaly detection under distribution mismatch, because they are developed with the assumption that labeled and unlabeled data come from the same distribution. Generated pseudo-labels are highly dependent on a small set of labeled data; thus, the trained semi-supervised models would be biased on the labeled data distribution. Transfer learning methods or the frameworks for distribution shifts may constitute alternatives (Pan & Yang, 2009; Yu et al., 2020; Raina et al., 2007) by treating source/target data as labeled/unlabeled data. However, these have not been effective with a small number of source (labeled) samples (as shown in Sec. 5).

Motivated by the common real-world scenarios, we tackle the distribution mismatch problem for semi-supervised anomaly detection which is critical but under-explored. We propose a novel semi-supervised anomaly detection framework, SPADE, that yields strong and robust performance even under distribution mismatch. The key aspects of SPADE can be summarized as below:

- **Semi-supervised learning to efficiently utilize unlabeled data:** Carefully-designed components enable robust semi-supervised learning, by combination of self-supervised and supervised learning stages.

| Frameworks | Description | Use of data | Examples |
|---|---|---|---|
| Supervised classification | Train supervised model with labeled data | L | MLP, RF, XGBoost |
| Negative supervised classification | Train supervised model while treating unlabeled data as normal data | L+U | MLP, RF, XGBoost |
| One-class classifier (OCC) | Train OCC only with labeled normal data | L(normal) | OC-SVM, GDE |
| Negative OCC | Train OCC while treating unlabeled data as normal data | L(normal)+U | OC-SVM, GDE |
| Unsupervised OCC | Train OCC with unlabeled data refinement | L(normal)+U | SRR (Yoon et al., 2022) |
| Semi-supervised learning | Train a predictive model via pseudo-labeling and representation learning | L+U | FixMatch (Sohn et al., 2020), VIME (Yoon et al., 2020) |
| Domain adaptation | Train a predictive model via domain-invariant representation learning | L+U | DANN (Ganin et al., 2016) |
| PU learning | Train a predictive model only with L (anomalous) + U via weighted ensemble learning | L(anomalous)+U | Elkanoto(Elkan & Noto, 2008), *BaggingPU*(Mordelet & Vert, 2014) |

Table 1: Conventional approaches to tackle anomaly detection with semi-supervised settings with distribution mismatch. (L: Labeled data, U: Unlabeled data, MLP: Multi-layer Perceptron, RF: Random Forest, GDE: Gaussian Distribution Estimator).

- **Data efficiency:** SPADE introduces a pseudo-labeling mechanism using an ensemble of OCCs and reduces the dependence on the labeled data as the predictors are trained with a small number of labeled and pseudo-labeled samples.

- **Robustness against distribution mismatch:** SPADE constructs more robust decision boundaries that can identify the pseudo-negative and pseudo-positive samples without overfitting to the labeled data, a critical aspect when there is distribution mismatch.

- **Selecting hyperparameters robustly:** We propose a novel approach using a partial matching method to pick the important hyperparameters without a validation set. This innovation is critical as conventional hyperparameter selection would rely on a validation set, which is often unavailable in real world with limited labeled data.

- **Strong results in real-world settings:** We show state-of-the-art semi-supervised anomaly detection performance of SPADE in multiple settings that represent common real-world scenarios. AUC improvements of SPADE can be up to 10.6% on tabular data and 3.6% on image data. We additionally focus on an important real-world machine learning challenge: fraud detection with distribution shifts over time due to the adversarial nature of the environment. We show that SPADE consistently outperforms alternatives.

## 2  Related Work

**Semi-supervised learning.** State-of-the-art methods (Sohn et al., 2020; Chen et al., 2020a; Grill et al., 2020) are developed under the assumption that both labeled and unlabeled samples come from the same distribution. They have pseudo-labeling approaches based on the consistency of label predictions with different augmentations. Such approaches are highly dependent on the small amount of labeled data. Thus, the bias from the labeled data would propagate to pseudo-labels of the unlabeled data, causing them to construct a biased predictive model if there is distribution mismatch between labeled and unlabeled data. Kim et al. (2020) tackles this in the setting where only the label priors are different. DeepSAD (Ruff et al., 2020) tackles semi-supervised anomaly detection problem while treating unlabeled samples as normal samples.

As a way of employing OCCs, SPADE differentiates from typical pseudo-labeling methods used in semi-supervised learning (Lee et al., 2013; Sohn et al., 2020) that require building binary classifiers to assign pseudo-labels. We argue that OCC-based pseudo-labeling is better-suited when there exists distribution mismatch between labeled and unlabeled data, a common pitfall for semi-supervised anomaly detection

applications, and more universally applicable (e.g., a binary classifier isn't available for PU settings). Yoon et al. (2022) also employs an ensemble of OCCs for fully-unsupervised settings. However, it only identifies pseudo-normal samples from unlabeled data and it needs prior knowledge on label distribution, which may not be available in practice (more details can be found in Appendix. A.4).

**Distribution mismatch.** Some recent works directly addressed the distribution mismatch between labeled and unlabeled data. (Chen et al., 2020b; Saito et al., 2021) assume that the distribution of labeled data and testing data are the same but the unlabeled data include additional out-of-distribution samples. Both papers focus on filtering out out-of-distribution samples from the unlabeled data to match the distribution between labeled and unlabeled data. On the other hand, in SPADE, the testing distribution is the union of the labeled and unlabeled distributions and the labeled data distribution is different from the testing distribution. Pang et al. (2019) assumes the existence of positively labeled samples which are included in the PU scenarios in SPADE. Pang et al. (2021) further assumes new anomaly types in unlabeled data, which is also addressed in this paper (see Sect. 5.1).

**Domain adaptation.** Various methods have been proposed to address the issue of the training distribution being different from the testing distribution (Long et al., 2016; Baktashmotlagh et al., 2013; Sun et al., 2019). These often focus on learning domain-invariant representations for better generalization to testing set with different distributions. If we assume that we have access to features of the test data (which is a common assumption in domain adaptation), we can consider the domain adaptation problem as a semi-supervised learning problem where training data are treated as labeled and test data are treated as unlabeled. However, with small amount of labeled data (less common in domain adaptation setting), the performance of the trained model on a small source data would be limited.

**Positive-Unlabeled (PU) learning.** An important special scenario is when we only have the positive samples as the labeled data, while unlabeled data include both positive and negative samples (Zhang & Zuo, 2008). In this setting, the labeled data distribution is clearly different from the unlabeled data, as a special case of semi-supervised anomaly detection with distribution mismatch. Related literature on PU learning is summarized in Bekker & Davis (2020). There are two commonly-used approaches: (i) two-stage models (He et al., 2018; Chaudhari & Shevade, 2012), where the first stage is discovering the *confident* negative labels and the second stage is training the supervised model using positive labels and *confident* negative labels; (ii) biased learning by treating all the unlabeled data as negative samples with class label noise (Liu et al., 2003; Sellamanickam et al., 2011). The shortcoming of (i) is excluding the possible positive samples from unlabeled data, whereas the shortcoming of (ii) is contamination of unlabeled data that affects model training. While being relevant, these are limited to the special case of PU setting, and sub-optimal when applied to the general semi-supervised settings.

## 3 Problem Formulation

We focus on the general semi-supervised anomaly detection problem with distribution mismatch. Consider the given labeled training data $\mathcal{D}^l = \{(\mathbf{x}_i^l, y_i^l)\}_{i=1}^{N_l}$ and unlabeled training data $\mathcal{D}^u = \{\mathbf{x}_j^u\}_{j=1}^{N_u}$. $\mathbf{x}^l \sim \mathcal{P}_X^l$ and $\mathbf{x}^u \sim \mathcal{P}_X^u$ are the feature vectors and $\mathcal{P}_X^l$ and $\mathcal{P}_X^u$ are corresponding feature distributions of the labeled and unlabeled data, respectively. For anomaly detection, the labels $y \in \mathcal{Y}$ are either normal (0) or anomalous (1) and there are far more normal examples than anomaly, i.e., $\mathbb{P}(y=0) \gg \mathbb{P}(y=1)$. Most semi-supervised methods assume that both labeled and unlabeled data come from the same distribution (i.e., $\mathcal{P}_X^l = \mathcal{P}_X^u$). In this work, we aren't limited by this assumption and allow the scenario of the distributions between labeled and unlabeled data to be different (i.e., $\mathcal{P}_X^l \neq \mathcal{P}_X^u$). We exemplify such scenarios in Fig. 1. For instance, if new anomaly types are only included in the unlabeled data, $\mathcal{P}_X^u$ would be different from $\mathcal{P}_X^l$. The labels $y$ are determined by the unknown function $f^* : \mathcal{X} \to \mathcal{Y}$ where $\mathbf{x}^l, \mathbf{x}^u \in \mathcal{X}$. Our main objective is to construct an anomaly detection model $f : \mathcal{X} \to \mathcal{Y}$ that can minimize the test loss $\mathcal{L}(f(x), y)$ in the union of $\mathcal{P}_X^l$ and $\mathcal{P}_X^u$. As a way of motivation, the conventional approaches to tackle this problem along with their limitations are summarized in Table. 1. All these are quantitatively compared with SPADE in Sec. 5. Further details can be found in Appendix A.

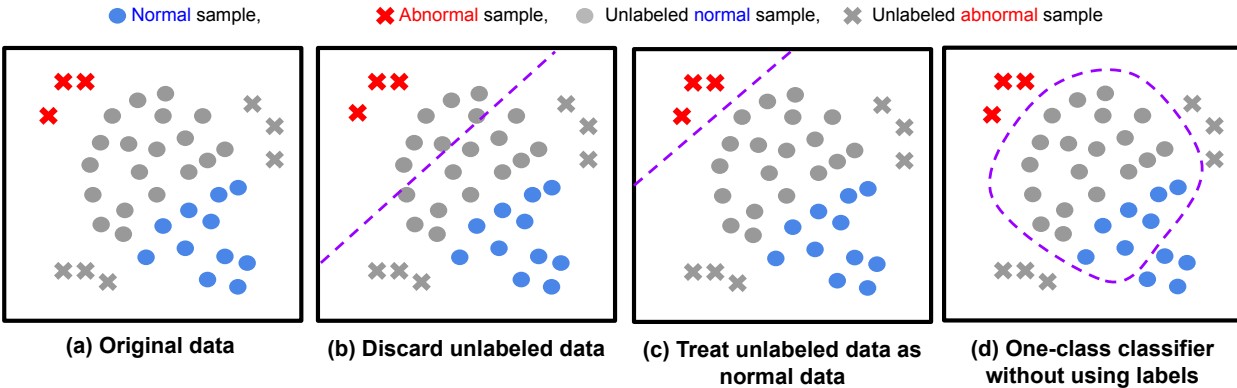

Figure 2: Examples in semi-supervised anomaly detection with distribution mismatch. (a) Original data distribution. Note that the labeled (color) and unlabeled (grey) data distributions are different; (b) Standard supervised learning approach only with labeled data; (c) Standard supervised learning approach after treating all the unlabeled data as normal samples; and (d) OCC without using labels. Purple line represents the decision boundary.

## 4 Proposed Method - SPADE

Sec. 4.1 first explains the design principles of SPADE, and then the implementation details are provided in the subsequent subsections. Sec. 4.2 introduces building blocks of the framework, Sec. 4.3 and 4.4 explain the details of the pseudo-labeler and Sec. 4.5 describes loss functions and optimization.

### 4.1 Desiderata

The core idea of our framework, Semi-supervised Pseudo-labeler Anomaly Detection with Ensembling (SPADE), is based on self-training, following recent advances in semi-supervised learning (Sohn et al., 2020; Chen et al., 2020a). We aim to train a binary classifier for normal and anomalous data by iteratively learning from labeled and pseudo-labeled data. As such, the key component is the pseudo-labeler to assign binary labels to unlabeled data. While it is common to use a trained binary classifier for pseudo-labeling (Lee et al., 2013; Sohn et al., 2020), we argue that it may be sub-optimal for anomaly detection with distribution shift as the decision boundaries of binary classifiers could be highly biased by the small labeled data. As shown in Fig. 2 (b, c), heavily relying on the labeled data or training with noisy labeled data would have a negative impact when labeled and unlabeled data distributions are mismatched. On the other hand, with OCCs (without using the labeled data at all), we can achieve quite reasonable decision boundaries (Fig. 2(d)) - still not perfect due to not using labeled information.

In SPADE, we incorporate these motivations and construct the pseudo-labeler in a way that it relies less on the labeled data. More specifically, when constructing the OCCs, SPADE excludes the positive labeled data to avoid overfitting to a small number of positive labeled data. In addition, SPADE uses the consensus approach on pseudo-labeling to significantly reduce the label noise in pseudo-labeled samples. As such, SPADE can generalize better to when there is a distribution mismatch.

### 4.2 Building blocks

Fig. 3 illustrates the four components of SPADE framework: (i) (data) encoder, (ii) predictor, (iii) pseudo-labeler, and (iv) projection head. First, the encoder: $h : \mathcal{X} \to \mathcal{H}$ maps the input features $\mathbf{x}$ into latent representations $\mathbf{r} = h(\mathbf{x})$. As the encoder, any neural network architecture can be employed – in our experiments, we use multi-layer perceptron (MLP) for tabular data and convolutional neural networks (CNNs) for image data. The predictor $q : \mathcal{H} \to \mathcal{Y}$ utilizes the learned representation $\mathbf{r}$ to output the anomaly scores $q(\mathbf{r})$. The anomaly score is determined by the encoder ($h$) and predictor ($q$) as follows: $q(h(\mathbf{x}))$. Pseudo-labeler and projection head help the encoder and predictor training. Pseudo-labeler $v : \mathcal{H} \to \{0, 1, -1\}$ determines

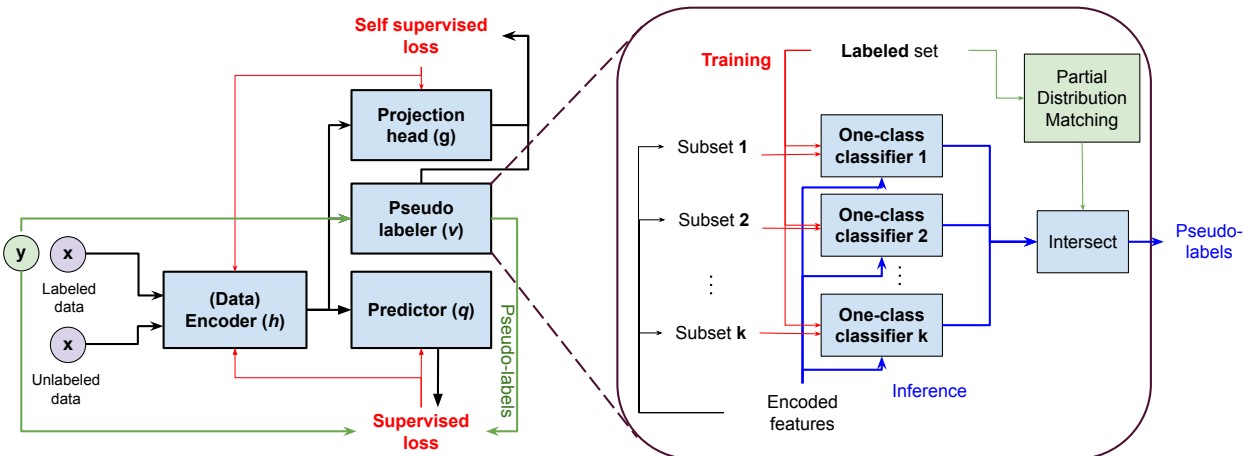

Figure 3: (Left) Block diagram of the proposed semi-supervised anomaly detection framework, SPADE. (Right) We zoom in the detailed block diagram of the proposed pseudo-labeler which is an ensemble of OCCs. Predictor is a binary classifier. Blue line represents the inference steps.

the pseudo-labels of the unlabeled data $\mathbf{x}^u$ using an ensemble of OCCs. $v(h(\mathbf{x}^u)) = 1/0/-1$ represents pseudo-anomalous/pseudo-normal/unlabeled. The predictor only utilizes the labeled data and unlabeled data with $v(h(\mathbf{x}^u)) = 1/0$ for training. Lastly, projection head $g : \mathcal{H} \rightarrow \mathcal{G}$ is the block to help representation learning of the encoder. Any representation learning method can be utilized, such as contrastive learning and pretext task predictions (such as masked autoencoder).

### 4.3 Pseudo-labeling via consensus

A major novel component of SPADE is the design of pseudo-labeler. The pseudo-labeler ($v$ in Fig. 3) is composed of an ensemble of $K$ OCCs ($o_1, o_2, ..., o_K$). Each OCC is trained with the negative labeled data ($\mathcal{D}_0^l$) and one of $K$ disjoint subsets of unlabeled data ($\mathcal{D}_1^u, \mathcal{D}_2^u, ..., \mathcal{D}_K^u$). $o_k(\mathbf{x})$ outputs the anomaly scores of $\mathbf{x}$. We assign the positive pseudo-labels (i.e. anomalous predictions) to unlabeled data samples if all OCCs agree on them: $v(h(\mathbf{x}^u)) = 1$ if $\prod_{k=1}^{K} \hat{y}_k^{pu} = 1$ where

$$\hat{y}_k^{pu} = \begin{cases} 1 & \text{if } o_k(h(\mathbf{x}^u)) > \eta_k^p \\ 0 & \text{otherwise} \end{cases} \tag{1}$$

Similarly, we assign a negative pseudo-label (i.e., normal) if all OCCs agree on negative pseudo-labels: $v(h(\mathbf{x}^u)) = 0$ if $\prod_{k=1}^{K} \hat{y}_k^{nu} = 1$ where

$$\hat{y}_k^{nu} = \begin{cases} 1 & \text{if } o_k(h(\mathbf{x}^u)) < \eta_k^n \\ 0 & \text{otherwise} \end{cases} \tag{2}$$

Unlabeled data without consensus are annotated as unknown: $v(h(\mathbf{x}^u)) = -1$ if $\prod_{k=1}^{K} \hat{y}_k^{pu} \times \hat{y}_k^{nu} = 0$.

### 4.4 Determining $\eta^p, \eta^n$ using partial matching

In SPADE framework, thresholds $\eta^p$ and $\eta^n$ are critical parameters. One option is considering them as user-defined hyper-parameters and determining them by the hyper-parameter optimization. However, hyper-parameter tuning requires extra validation data which should come from labeled training set (same impacts as reducing the number of labeled samples in training data which is critical in semi-supervised setting). Instead, we propose to learn these parameters without sacrificing the labeled data for validation. We propose adapting the partial matching method (Christoffel et al., 2016), which has been developed to estimate the marginal distribution of unlabeled data by matching the distribution to the known one-class (either positive

or negative) distribution. The underlying intuition is that normal samples are closer to other normal samples, and anomalous samples are closer to other anomalous samples. In our case, we match the distribution of anomaly scores of the positive labeled data to that of unlabeled data to estimate their marginal distribution and determine $\eta^p$ accordingly. The same is applied to determine $\eta^n$ using negative labeled data. Formulations for $\eta^p$ and $\eta^n$ are given in Eqs. 3 and 4 below:

$$\eta_k^p = \arg\min_{\eta} D_w(\{o_k(h(\mathbf{x}^l))|y^l=1\}, \{o_k(h(\mathbf{x}^u))>\eta\}) \tag{3}$$

$$\eta_k^n = \arg\min_{\eta} D_w(\{o_k(h(\mathbf{x}^l))|y^l=0\}, \{o_k(h(\mathbf{x}^u))<\eta\}) \tag{4}$$

where $D_w$ is the Wasserstein distance between two distributions. That is, we determine the subsets of the unlabeled data for pseudo-labeling whose Wasserstein distance from labeled data is minimum. More specifically, the outputs of $o_k$ are the one-dimensional anomaly scores and we compute the Wasserstein distance between two one-dimensional anomaly scores. Wasserstein distance between two one-dimensional vectors can be computed as the integral of the cumulative distribution function differences.

In some semi-supervised settings such as PU and NU, only one-class of labeled samples are available. In that case, we employ Otsu's method (Otsu, 1979) to identify the threshold of the class without labeled samples. With Otsu's method, we can determine the threshold that minimizes intra-class anomaly score variances in an unsupervised way. More specifically, Otsu's method is applied to one-dimensional anomaly scores. For all unlabeled samples, we extract one-dimensional anomaly scores from the trained OCCs. Then, we find the threshold that minimizes the intra-class variances of two subgroups (splitted by the threshold) of anomaly scores. In PU setting, we set $\eta^p$ using Eq. 3 and $\eta^n$ using Otsu's method.

### 4.5 Loss functions and optimization

We train the anomaly detection model $q(h(\cdot))$ using three loss functions: (i) binary cross entropy (BCE) on labeled and (ii) BCE on pseudo-labeled data, and (iii) self-supervised loss on the entire data. The self-supervised module $g$ (e.g., decoder for reconstruction loss, MLP projection head for contrastive loss) is jointly trained with an auxiliary self-supervised loss.

Next, we describe the loss formulations. The BCE loss on the labeled data is proposed as:

$$\mathcal{L}_{Y^l} = \mathbb{E}\big[\mathcal{L}_{BCE}(q(h(\mathbf{x}^l)), y^l)\big],$$

and the BCE loss on pseudo-labeled data as:

$$\mathcal{L}_{Y^u} = \mathbb{E}\big[\mathcal{L}_{BCE}(q(h(\mathbf{x}^u)), v(h(\mathbf{x}^u))) \times \mathbb{1}\{v^u \in \{0,1\}\}\big].$$

Here, instead of subsampling unlabeled data with known pseudo-labels, we assign a binary weight ($\mathbb{1}\{v^u \in \{0,1\}\}$) to each unlabeled sample so that the loss contribution from pseudo-labeled data can be controlled based on the model quality.

To improve the quality of the encoder ($h$), we utilize auxiliary self-supervised losses with various pretext tasks depending on application domain. This may include the reconstruction objective:

$$\mathcal{L}_R = \mathbb{E}\big[\mathcal{L}_{MSE}(\mathbf{x}, g(h(\mathbf{x})))\big],$$

or more specific objectives to data type, such as contrastive learning (Chen et al., 2020a) and CutPaste (Li et al., 2021) for image.

Overall, the encoder ($h$), predictor ($q$), and the self-supervised module ($g$) are trained by solving the following optimization problem:

$$h^*, g^*, q^* = \arg\min_{h,g,q} \big[\mathcal{L}_{Y^l} + \alpha\mathcal{L}_{Y^u} + \beta\mathcal{L}_R\big], \tag{5}$$

where $\alpha, \beta$ are hyper-parameters (we set both $\alpha$ and $\beta$ as 1.0 for the experiments). Training loss is used for the convergence criteria – if the training loss is converged (if no improvement is observed in the loss for 5 epochs), we treat that the models are converged as well. Note that the pseudo-labeler also converges during training, often faster. The overall pseudo-code can be found in Alg. 1.

---

**Algorithm 1** Semi-supervised Pseudo-labeler Anomaly Detection with Ensembling (SPADE).

---

**Input**: Labeled / unlabeled training data $\mathcal{D}^l$ / $\mathcal{D}^u$
**Output**: Trained encoder ($h$), predictor ($q$)

1: **Initialize** $g, h, q$.
2: **Set** positively / negatively labeled data $\mathcal{D}_1^l, \mathcal{D}_0^l$
3: **while** $g, h, q$ not converged **do**
4:     $v$=PSEUDO-LABELER($\mathcal{D}_1^l, \mathcal{D}_0^l, \mathcal{D}^u, h$)
5:     Update $g, h, q$ using Eq. 5.
6: **end while**
7:
8: **function** PSEUDO-LABELER($\mathcal{D}_1^l, \mathcal{D}_0^l, \mathcal{D}^u, h$)
9:     Divide $\mathcal{D}^u$ into $K$ disjoint subsets $\{\mathcal{D}_k^u\}_{k=1}^K$
10:     **for** k=1:K **do**
11:         Train OCC models $o_k$ on $\mathcal{D}_k^u \cup \mathcal{D}_0^l$
12:         Set $\eta_k^p/\eta_k^p$ using partial matching with $\mathcal{D}_1^l, \mathcal{D}_0^l$ using Eqs. 3 and 4.
13:     **end for**
14:     Build pseudo-labeler $v$ following Eqs. 1 and 2.
15:     **Return** pseudo-labeler $v$.
16: **end function**

---

## 5  Experiments

We conduct extensive experiments to highlight the benefits of the proposed method, SPADE, in various practical settings of semi-supervised learning with distribution mismatch. We consider multiple anomaly detection datasets for image and tabular data types. As image data, we use MVTec anomaly detection (Bergmann et al., 2019) and Magnetic tile datasets (Huang et al., 2020). As tabular data, we use Covertype, Thyroid, and Drug datasets (see Appendix for detailed data description). In Sec. 5.4, we further utilize two real-world fraud detection datasets (Kaggle credit and Xente) to evaluate the performance of SPADE.

In all experiments, unless the dataset comes with its own train and test split, we randomly divide the dataset into disjoint train and test data. Then, we further divide the training data into disjoint labeled and unlabeled data. Note that we construct labeled and unlabeled data such that they come from different distributions (more details can be found in the following subsections). We run 5 independent experiments and report average values (standard deviations can be found in Appendix C). We use AUC as the evaluation metric. More experimental details (on model architectures, training settings, and pseudo-labelers) are provided in Appendix B. Computational complexity analyses can be found in Appendix B.7.

We compare SPADE to baselines from Table 1. Note that not all baselines are applicable to every scenario. More specifically, we use Gaussian Distribution Estimator (GDE) for both OCC (only using the negatively labeled data) and Negative OCC (only excluding the positively labeled data). Note that GDE performs the best in comparison to the alternatives in our experiments (including isolation forests, OC-SVM). We use SRR (Yoon et al., 2022) as the unsupervised OCC baseline and Random Forest as the supervised (only using the labeled data) and negative supervised (treat unlabeled data as negative) baselines. For image data, FixMatch is used instead of VIME as the semi-supervised baseline. We use CutPaste (Li et al., 2021) as the baseline architecture for Negative OCC, Unsupervised OCC, and SPADE for MVTec and Magnetic datasets.

### 5.1  New types of anomalies

Anomalies can evolve over time in many applications. For fraud detection, criminals might invent new fraudulent approaches to trick the existing systems; or for manufacturing, modified process might yield different defects that have been never met before. Therefore, labeled data can get out-dated and newly-gathered unlabeled data can come from different distributions. To mimic such scenarios, we construct datasets with multiple anomaly types. Among multiple anomaly types, we provide subsets of the anomaly types (and

normal samples) as the labeled data. It means that other anomaly types only appear in the unlabeled data. Detailed experimental settings can be found in Appendix. B.2.

| Datasets | Thyroid | | | Drug | | | Covertype | | |
|---|---|---|---|---|---|---|---|---|---|
| Metrics (AUC) | Overall | Given | Missed | Overall | Given | Missed | Overall | Given | Missed |
| Supervised | 0.815 | 0.996 | 0.741 | 0.818 | 0.810 | 0.833 | 0.858 | **0.988** | 0.693 |
| Negative Supervised | 0.622 | 0.837 | 0.533 | 0.676 | 0.670 | 0.685 | 0.761 | 0.881 | 0.610 |
| OCC | 0.711 | 0.876 | 0.643 | 0.741 | 0.727 | 0.765 | 0.897 | 0.910 | 0.880 |
| Negative OCC | 0.446 | 0.637 | 0.367 | 0.731 | 0.700 | 0.780 | 0.825 | 0.832 | 0.815 |
| Unsupervised OCC | 0.429 | 0.612 | 0.353 | 0.769 | 0.747 | 0.803 | 0.843 | 0.853 | 0.831 |
| VIME | 0.592 | 0.724 | 0.538 | 0.792 | 0.777 | 0.820 | 0.837 | 0.967 | 0.672 |
| DANN | 0.725 | 0.876 | 0.662 | 0.744 | 0.730 | 0.768 | 0.791 | 0.979 | 0.552 |
| SPADE (Ours) | **0.921** | **0.997** | **0.891** | **0.837** | **0.831** | **0.849** | **0.928** | 0.957 | **0.892** |

Table 2: Experimental results with new types of anomalies scenario in terms of Overall / Given / Not given (Missed) AUC. Overall/Given/Missed: Put all/given/missed anomaly types and normal samples in the test set for evaluation.

Tables 2 and 4 (left) show that SPADE achieves consistently and significantly better performance in all 3 metrics (overall, given, and missed AUC), demonstrating its generalizability to unseen anomalies. On the other hand, supervised and semi-supervised (VIME and FixMatch) methods remain highly biased towards given anomalies and generalize poorly to new types of anomalies. Compared to the best baseline, SPADE improves overall AUC by 0.106, 0.015, and 0.031 on the three tabular datasets.

Each baseline has its own limitations. Supervised classifiers cannot utilize unlabeled data at all, and negative supervised classifier suffers from contaminated labeled data for training the predictive model. OCC models are suboptimal as they cannot utilize the anomalous label information. Semi-supervised learning baselines suffer from distribution mismatch between labeled and unlabeled data. For domain adaptation baseline, it shows poor performances with a small number of source samples.

### 5.2 Labeling based on the 'easiness' of samples

The difficulty for human labeling may differ across different samples – while some samples are easy to label, some samples can be misleadingly difficult to humans because they appear differently from the known cases. To simulate this scenario, we focus on an experiment where the labeled data only includes easy-to-label samples while hard-to-label samples are included in the unlabeled dataset. To this end, we train logistic regression using the entire training data, and gather the labeled samples where confidence of the trained logistic regression outputs are larger than a certain threshold and the predictions are correct. Details can be found in Appendix. B.3.

| Datasets | Thyroid | Drug | Covertype |
|---|---|---|---|
| Supervised | 0.805 | **0.848** | 0.878 |
| Negative Supervised | 0.626 | 0.701 | 0.599 |
| OCC | 0.787 | 0.838 | 0.888 |
| Negative OCC | 0.464 | 0.741 | 0.826 |
| Unsupervised OCC | 0.484 | 0.786 | 0.846 |
| VIME | 0.728 | 0.849 | 0.843 |
| DANN | 0.731 | 0.754 | 0.835 |
| SPADE (Ours) | **0.833** | 0.846 | **0.892** |

Table 3: Experimental results with labeling based on the 'easiness' of samples in terms of overall AUC.

| Scenarios | New anomalies | | Easiness | |
|---|---|---|---|---|
| Datasets | MVTec | Magnetic | MVTec | Magnetic |
| Supervised | 84.3 | 82.3 | 90.9 | 81.7 |
| Negative Supervised | 76.5 | 63.5 | 79.2 | 59.3 |
| Negative OCC | 81.3 | 69.0 | 87.6 | 70.1 |
| Unsupervised OCC | 85.4 | 72.2 | 88.4 | 73.1 |
| FixMatch | 81.4 | 69.1 | 83.5 | 70.8 |
| SPADE (Ours) | **87.9** | **85.2** | **92.1** | **83.9** |

Table 4: Experimental results on image domain with (left) new types of anomalies, (right) labeling based on easiness scenarios in terms of overall AUC.

Tables 3 and 4 (right) show that SPADE achieves superior or similar anomaly detection performances compared to the best alternative. This constitutes a great potential in reducing human labeling costs by allowing the labelers to skip samples if they would take too long to correctly label. The experimental results with the opposite setting (only labeling the high-risk samples) can also be found in Appendix D.1.

### 5.3 PU (Positive & Unlabeled) learning

With only positive samples as the labeled data and all other samples being unlabeled, i.e. the positive and unlabeled (PU) settings, the distributions between labeled (only positive samples) and unlabeled (both positive and negative samples) would be different. We use the same experimental settings with the 'new types of anomalies' scenario except additionally excluding normal samples from the labeled data, to represent PU setting. Detailed experimental settings can be found in Appendix. B.4.

| Datasets | Thyroid | | | Drug | | | Covertype | | |
|---|---|---|---|---|---|---|---|---|---|
| Metrics (AUC) | Overall | Given | Missed | Overall | Given | Missed | Overall | Given | Missed |
| Negative Supervised | 0.786 | **0.997** | 0.698 | 0.839 | 0.839 | **0.840** | 0.846 | **0.996** | 0.657 |
| Negative OCC | 0.470 | 0.695 | 0.377 | 0.739 | 0.709 | 0.787 | 0.849 | 0.864 | 0.831 |
| Unsupervised OCC | 0.519 | 0.707 | 0.441 | 0.771 | 0.748 | 0.809 | 0.863 | 0.880 | 0.842 |
| Weighted Elkanoto (Elkan & Noto, 2008) | 0.772 | 0.934 | 0.705 | 0.711 | 0.714 | 0.706 | 0.699 | 0.917 | 0.422 |
| BaggingPU (Mordelet & Vert, 2014) | 0.787 | 0.964 | 0.714 | 0.734 | 0.740 | 0.724 | 0.726 | 0.907 | 0.497 |
| SPADE (Ours) | **0.929** | 0.996 | **0.901** | **0.840** | **0.842** | 0.837 | **0.896** | 0.940 | **0.839** |

Table 5: Experimental results on PU settings on 3 tabular datasets in AUC of overall/given/missed (not given). Due to the absence of negatively-labeled samples, Supervised, OCC, semi-supervised, and domain adaptation baselines are excluded. Instead, two PU baselines are included.

Table 5 compares the performances of the proposed method (SPADE) in PU settings on multiple tabular datasets. SPADE generalizes much better and outperforms all other alternatives with significantly better AUC in missed (not given) anomaly types. Note that PU baselines severely suffer from distribution mismatches when new types of anomalies are included in the unlabeled data.

### 5.4 Time-varying distributions: real-world fraud detection

We evaluate the proposed framework with two real-world fraud detection datasets: (i) Kaggle credit card fraud[1] (0.17% anomaly ratio with total 284807 samples), (ii) Xente fraud detection[2] (0.20% anomaly ratio with total 95662 samples). For these tasks, anomalies are evolving (i.e., their distributions are changing over time) (Grover et al., 2022). To catch the evolving anomalies, we need to keep labeling for new anomalies

---

[1] https://www.kaggle.com/datasets/mlg-ulb/creditcardfraud
[2] https://zindi.africa/competitions/xente-fraud-detection-challenge/data

and retrain the anomaly detection model. However, labeling is costly and time consuming. Even without additional labeling, SPADE can improve the anomaly detection performance using both labeled data and newly-gathered unlabeled data.

| Datasets | Kaggle Credit Fraud | | Xente Fraud | |
|---|---|---|---|---|
| Labeling ratio | 5% | 10% | 10% | 20% |
| Supervised | 0.975 | 0.977 | 0.906 | 0.925 |
| Negative Supervised | 0.971 | 0.976 | 0.909 | 0.918 |
| OCC | 0.717 | 0.803 | 0.891 | 0.920 |
| Negative OCC | 0.838 | 0.835 | 0.608 | 0.630 |
| Unsupervised OCC | 0.897 | 0.897 | 0.806 | 0.912 |
| VIME | 0.941 | 0.943 | 0.859 | 0.893 |
| DANN | 0.921 | 0.922 | 0.798 | 0.822 |
| SPADE (Ours) | **0.982** | **0.983** | **0.920** | **0.931** |

Table 6: Experimental results on two real-world fraud detection datasets in terms of overall AUC.

In our experiments, we split the train and test data based on the measurement time. Latest samples are included in the testing data (50%) and early acquired data is included in the training data (50%). We further divide the training data as labeled and unlabeled data. Early acquired data are included in the labeled data (5%-20%), while later acquired data are included in the unlabeled data (80%-95%). We use AUC as the anomaly detection metric. As shown in Table. 6, SPADE consistently outperforms alternatives for different labeling ratio values on both datasets, taking advantage of the unlabeled data and showing robustness to evolving distributions.

## 6 Discussions

**Accuracy of the pseudo-labels.** SPADE is based on the proposed pseudo-labeling mechanism. The accuracy of the pseudo-labeler is highly related to the robustness of semi-supervised anomaly detection. We analyze the accuracy (in precision) of the pseudo-labels vs. anomaly score percentiles for both normal and anomalous samples.

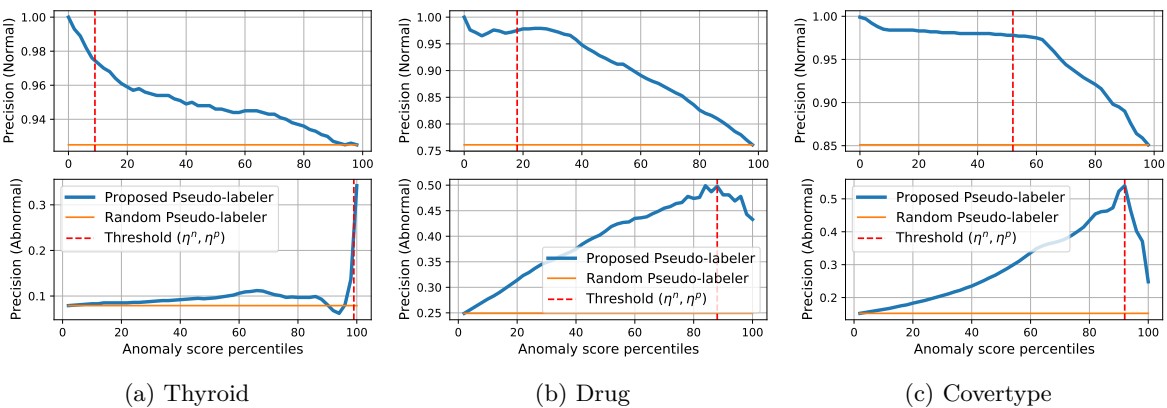

(a) Thyroid  (b) Drug  (c) Covertype

Figure 4: Precision for pseudo-labelers across anomaly score percentiles on 3 tabular datasets with new types of anomalies. $\eta^n, \eta^p$ represents the discovered threshold for normal and anomalous pseudo-labels by partial matching in percentile.

Fig. 4 shows that the proposed pseudo-labeler achieves fairly robust pseudo-labeling for normal samples. On the other hand, for anomalous samples, the precision of pseudo-labeling gets high typically when the

anomaly scores are higher than 80%, however we observe drop in precision in some cases, which we attribute to imperfect OCC fitting. While this underlines the room for improvement for pseudo-labeling, due to the robustness of partial matching, the impact of imperfect precision on anomaly detection performance is low. Note that our partial matching algorithm catches this threshold fairly accurately to make pseudo-labels robust without any threshold parameter tuning.

**Ablation studies.** SPADE consists of multiple components and understanding the impact of each component is of high importance. In Table. 7, we demonstrate the performance impacts of 5 different components in SPADE on the Thyroid data with the setting of new anomaly types. All components of the SPADE are observed to contribute to the robust anomaly detection performance considerably. Self-supervised learning component contributes to 0.018 AUC improvements of SPADE framework. In addition, with majority votes instead of unanimous votes for pseudo-labeling, the performance of SPADE is degraded by 0.024 AUC. Additional ablation studies on other datasets can be found in Appendix D.2 and D.3.

$\alpha$ is a critical hyper-parameter of SPADE determining the importance of pseudo-label loss in comparison to given labeled data. We analyze the impact of this hyper-parameter in Fig. 5. With $\alpha = 0$, the performance is much worse than $\alpha > 0$ on Thyroid (0.08 lower AUC) and on Covertype (0.06 lower AUC) datasets. This underlines the impact of utilizing the unlabeled data. In addition, the performances are similar across different $\alpha > 0$, demonstrating that SPADE isn't sensitive to the hyper-parameter $\alpha$. Note that, all the experiments in Sec. 5 use $\alpha = 1$ as the default value.

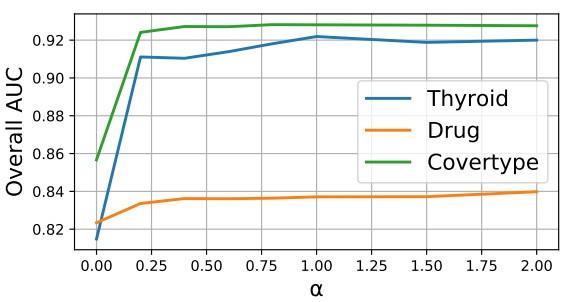

| Variants | Overall AUC |
|---|---|
| (i) No partial matching | 0.898 |
| (ii) No ensemble | 0.894 |
| (iii) $\beta = 0$ (No self-supervised) | 0.903 |
| (iv) No normal samples | 0.901 |
| (v) Majority vote | 0.897 |
| **SPADE** | **0.921** |

Table 7: Ablation studies on Thyroid dataset in new anomaly settings: (i) without partial matching, (ii) without an ensemble of OCC, (iii) with $\beta = 0$ (No self-supervised learning), (iv) without normal samples for pseudo-labeler training, (v) majority vote instead of unanimous votes for pseudo-labeling.

Figure 5: Overall AUC across different values of $\alpha$ using three tabular datasets. ($\alpha = 0$ represents SPADE without utilizing pseudo-labels.)

# 7 Conclusions

Semi-supervised anomaly detection is a highly-important challenge in practice – in many scenarios, we cannot assume that the distributions of labeled and unlabeled samples are the same. In this paper, we focus on this and demonstrate the underperformance of standard frameworks in this setting. We propose a novel framework, SPADE, which introduces a novel pseudo-labeling mechanism using an ensemble of OCCs and a judicious way of combining supervised and self-supervised learning. In addition, our framework involves a novel approach to pick hyperparameters without a validation set, a crucial component for data-efficient anomaly detection. Overall, we show that SPADE consistently outperforms the alternatives in various scenarios – AUC improvements with SPADE can be up to 10.6% on tabular data and 3.6% on image data. In addition to anomaly detection, future work can extend this semi-supervised framework to multi-class classification or regression with distribution mismatch.

# Acknowledgement

We gratefully acknowledge the contributions of Kyle Ziegler and Nate Yoder.

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

# Appendix

## A   Details of the conventional solutions

### A.1   Standard supervised learning

The most straightforward approach is applying the standard supervised learning framework. We can construct the supervised model $g_{sup}$ only with the labeled data $\mathcal{D}^l$ as follows.

$$g_{sup} = \arg\max_g \sum_{i=1}^{N_l} \mathcal{L}(g(x_i^l), y_i^l)$$

However, in this case, we cannot benefit from the unlabeled data $\mathcal{D}^u$ which can be beneficial for further boosting the performance with various semi-supervised learning framework. Also, the training data distribution $\mathcal{X}^l$ is different from the testing distribution $\mathcal{X}$ which can negatively impact on the test performance. We may treat all the unlabeled data as normal samples and apply the supervised learning framework ($g_{sup+}$) as follows:

$$g_{sup+} = \arg\max_g \Big[ \frac{1}{N_l} \sum_{i=1}^{N_l} \mathcal{L}(g(x_i^l), y_i^l) + \frac{1}{N_u} \sum_{j=1}^{N_u} \mathcal{L}(g(x_j^u), 0) \Big].$$

However, in this case, labeled normal samples are contaminated.

### A.2   Standard one-class classifiers (OCCs)

OCCs are one of the most promising methods to tackle the anomaly detection problem. Instead of using incomplete anomaly labels, we can only utilize the labeled normal samples $\mathcal{D}_0^l = \{(x_j, y_j) \in \mathcal{D}^l | y_j = 0\}$ to construct the OCC ($g_{one}$). However, in this case, we need to drop all labeled abnormal samples and unlabeled samples which may include quite critical information. We can include the unlabeled data ($\mathcal{D}^u$) to construct another OCC ($g_{one+}$) such as SRR (Yoon et al., 2022). However, it still cannot utilize the labeled abnormal samples.

### A.3   Semi-supervised learning

With both labeled and unlabeled data, we usually prioritize to apply semi-supervised learning approaches. We can utilize the semi-supervised learning framework to construct the anomaly detection model as follows.

$$g_{semi} = \arg\max_g \sum_{i=1}^{N_l} \mathcal{L}(g(x_i^l), y_i^l) + \lambda \sum_{j=1}^{N_u} \mathcal{L}^u(g(x_j^u))$$

Most semi-supervised learning frameworks assume that the labeled data $\mathcal{D}^l$ and unlabeled data $\mathcal{D}^u$ come from the same distribution. However, this assumption does not hold in our problem formulation. Thus, possibly-biased labeled data distribution can negatively affect on the trained semi-supervised model.

### A.4   Detailed comparison with SRR (Yoon et al., 2022)

SPADE has some resemblance with the SRR paper (Yoon et al., 2022). However, there are clear differences between SPADE and SRR. First, the problem setting is different. One of the biggest novelties of SPADE is tackling an important but under-explored problem: semi-supervised learning with distribution mismatch (e.g., common labeling bias). This has not been discussed in SRR which focused on only the fully unsupervised setting. Extension from fully unsupervised to general semi-supervised setting is not straightforward. Second, the approach to utilize the positive and negative samples is not discussed in SRR, which is critical in SPADE. We should consider how we utilize the normal samples for improving the pseudo-labeler training (please see the ablation studies in Table 6) and how we utilize the labeled samples for determining the thresholds - Line 4 and 5 in Algorithm 1. Third, SPADE can automatically determine the thresholds parameters without true anomaly ratios or validation set by the proposed partial matching.

# B    Detailed experimental settings

## B.1    Convert multi-class datasets into anomaly detection datasets

- For Thyroid data[3], there are 3 classes. The class distributions are (1: 2.47%, 2: 5.06%, 3: 92.47%). We treat label 3 as the normal samples and label 1 and 2 as the abnormal samples. We use the pre-defined training and testing dataset division.

- For Drug data[4], there are 7 classes. The class distributions are (1: 75.27%, 2: 2.02%, 3: 4.56%, 4: 8.28%, 5: 3.29%, 6: 2.12%, 7: 4.46%). We treat label 1 as the normal samples and all the other labels as the abnormal samples. We divide the entire dataset into training (50%) and testing (50%).

- For Covertype data[5], there are 7 classes. The class distributions are (1: 36.55%, 2: 48.75%, 3: 6.14%, 4: 0.47%, 5: 1.64%, 6: 2.94%, 7: 3.50%). We treat label 1 and 2 as the normal samples and all the other labels as the abnormal samples. We divide the entire dataset into training (50%) and testing (50%).

- For MVTec data (Bergmann et al., 2021)[6], different categories (15 categories) have different number of anomaly types. We set type 0 as the normal samples and all the other types as abnormal samples. Note that we first mix given training and testing data and divide them into training (80%) and testing (20%) to make the same abnormal ratio between training and testing sets.

- For Magnetic Tile dataset (Huang et al., 2020)[7], there are 6 types of samples: free, blowhole, crack, break, fray, and uneven. We set the free type as the normal, and the other 5 types as anomalies. We mix given training and testing data and divide them into training (80%) and testing (20%) to have the same abnormal ratio between training and testing sets.

## B.2    Detailed experimental settings in Scenario 1: New types of anomalies

On each of the 5 datasets that we used in this paper, there are multiple types of anomalies. In such scenarios, we only provide a subset of anomaly types as the labeled data and the rest of the anomaly types as the unlabeled data. The below explains which types of anomalies are provided as the labeled data for each dataset:

- For Thyroid data, we provide label 1 anomaly type to the labeled data (label 2 anomaly type is only in the unlabeled data).

- For Drug data, we provide label 2,3,4 as anomaly types to the labeled data (label 5, 6, 7 anomaly types are only in the unlabeled data).

- For Covertype data, we provide label 3, 4, 5 as anomaly types to the labeled data (label 6, 7 anomaly types are only in the unlabeled data).

- For MVTec and Magnetic tile datasets, different categories have different number of anomaly types. We provide the odd types as anomaly types to the labeled data. All the even types of anomalies are included in the unlabeled data.

Note that we only provide 5% of the data as labeled data for tabular datasets and 20% for image datasets, for the scenario of new types of anomalies.

---

[3] https://archive.ics.uci.edu/ml/datasets/thyroid+disease
[4] https://archive.ics.uci.edu/ml/datasets/Drug+consumption+%28quantified%29
[5] https://archive.ics.uci.edu/ml/datasets/covertype
[6] https://www.mvtec.com/company/research/datasets/mvtec-ad
[7] https://github.com/abin24/Magnetic-tile-defect-datasets.

### B.3 Detailed experimental settings in Scenario 2: Labeling based on the easiness of samples

To identify the easiness of the samples, we train a logistic regression model using the entire training data, and we gather the labeled samples where confidence of the trained logistic regression model outputs are larger than a certain threshold and the predictions are correct. To balance the labeled data in both normal and abnormal samples, we select top 10% confidences (from trained logistic regression) of each class as the labeled data for tabular datasets (20% confidence for image datasets). The rest of the samples are treated as the unlabeled samples.

### B.4 Detailed experimental settings in Scenario 3: PU learning

The experimental settings in PU settings are the same with scenario 1 (new types of anomaly) except the following points:

- We exclude all the normal samples from the labeled data to make the experiments in PU setting.

- We provide 50% of the given anomaly types as the labeled data. However, the number of labeled data is less than Scenario 1 because we exclude all the normal samples from the labeled data.

### B.5 Details on model architecture and training

For image data, we use ResNet-18 as the base network architecture. For representation learning, we incorporate CutPaste (Li et al., 2021) for MVTec and Magnetic Tile datasets. We follow all the training details in (Li et al., 2021) (including all the hyper-parameters).

For tabular data, we use two-layer perceptron as the base network architecture where the hidden dimensions is the half of the original feature dimensions. Pseudo-labelers consist of 5 Gaussian Distribution Estimator (GDE) based OCCs. For each epoch, we update the ensemble of OCCs for the pseudo-labels and further training the data encoder, projection head, and predictor. We set $\alpha = 1$ and $\beta = 1$ for all the experiments except the ablation studies. We use the training loss as the convergence criteria. More specifically, if the training loss does not improve for 5 epochs, we stop model training.

train OCCs, we only need data from a single class - we do not need label information. For pseudo-labeler of SPADE, we treat the negative labeled data and one of $K$ disjoint subsets of unlabeled data as the one-class data to train the OCCs. We use Gaussian Distribution Estimator (GDE) which utilizes one-class training data (negative labeled data and subsets of unlabeled data) to estimate the density function with maximum likelihood objective for the distribution assumption as the loss function. At inference, the likelihood outputs of GDE for each sample are used as the anomaly scores.

### B.6 Baselines

We compare SPADE with various baselines in different settings. Below describes the details of the baselines used in this paper:

- Gaussian Distribution Estimator (GDE) for both OCC (only using the negatively labeled data) and Negative OCC (only excluding the positively labeled data)[8].

- Random Forests for the supervised (only using the labeled data) and negative supervised (treat all the unlabeled data as negative)[9]

- VIME[10] for the tabular semi-supervised learning baseline and FixMatch[11] for the image semi-supervised learning baseline.

- Domain Adversarial Neural Network (DANN) for the domain adaptation baseline[12].

---

[8]https://scikit-learn.org/stable/modules/generated/sklearn.mixture.GaussianMixture.html

[9]https://scikit-learn.org/stable/modules/generated/sklearn.ensemble.RandomForestClassifier.html

[10]https://github.com/jsyoon0823/VIME

[11]https://github.com/google-research/fixmatch

[12]https://github.com/pumpikano/tf-dann

- Weighted Elkanoto[13] and BaggingPU[14] for PU learning baselines.

- CutPaste for the base architecture of image domain anomaly detection[15].

### B.7 Computational complexity analyses

All the experiments are done on a single V100 GPU. For tabular datasets, training and inference need at most 1 hour per each experiment (with the largest dataset, Covertype). For image dataset, training and inference need at most 4 hours per each experiment, mostly due to the representation learning with CutPaste. Note that the pseudo-labeler (an ensemble of OCCs) is re-trained per an epoch (not per an iteration) and we use shallow OCCs (GDE) for the ensemble to further reduce the computational complexity.

## C  Standard deviations of the experiment results

In this section, we report the standard deviations of the experimental results described in the main manuscript across 5 independent runs.

| Datasets | Thyroid | | | Drug | | | Covertype | | |
|---|---|---|---|---|---|---|---|---|---|
| Metrics (AUC) | Overall | Given | Missed | Overall | Given | Missed | Overall | Given | Missed |
| Supervised | 0.051 | 0.003 | 0.076 | 0.028 | 0.031 | 0.031 | 0.003 | 0.001 | 0.008 |
| Negative Supervised | 0.037 | 0.094 | 0.025 | 0.058 | 0.062 | 0.055 | 0.003 | 0.004 | 0.004 |
| OCC | 0.094 | 0.074 | 0.108 | 0.062 | 0.071 | 0.052 | 0.001 | 0.001 | 0.001 |
| Negative OCC | 0.002 | 0.006 | 0.001 | 0.020 | 0.022 | 0.021 | 0.001 | 0.002 | 0.001 |
| Unsupervised OCC | 0.017 | 0.034 | 0.010 | 0.013 | 0.016 | 0.018 | 0.001 | 0.002 | 0.001 |
| VIME | 0.068 | 0.064 | 0.072 | 0.075 | 0.080 | 0.067 | 0.014 | 0.001 | 0.032 |
| DANN | 0.063 | 0.075 | 0.061 | 0.084 | 0.083 | 0.088 | 0.010 | 0.001 | 0.022 |
| SPADE (Ours) | 0.029 | 0.001 | 0.041 | 0.024 | 0.026 | 0.026 | 0.001 | 0.001 | 0.002 |

Table 8: Standard deviations of experiments with new types of anomalies scenario in terms of Overall / Given / Not given (Missed) AUC. Overall/Given/Missed: Put all/given/missed anomaly types and normal samples in the test set for evaluation.

| Datasets | Thyroid | Drug | Covertype |
|---|---|---|---|
| Supervised | 0.013 | 0.009 | 0.002 |
| Negative Supervised | 0.010 | 0.033 | 0.002 |
| OCC | 0.031 | 0.016 | 0.001 |
| Negative OCC | 0.004 | 0.020 | 0.002 |
| Unsupervised OCC | 0.015 | 0.016 | 0.001 |
| VIME | 0.033 | 0.015 | 0.017 |
| DANN | 0.045 | 0.037 | 0.018 |
| SPADE (Ours) | 0.021 | 0.014 | 0.001 |

Table 9: Standard deviations of experiments with labeling based on the 'easiness' of samples in terms of overall AUC.

---

[13]https://pulearn.github.io/pulearn/doc/pulearn/index.html#weighted-elkanoto
[14]https://pulearn.github.io/pulearn/doc/pulearn/index.html#bagging-based-pu-learning
[15]https://github.com/Runinho/pytorch-cutpaste

| Scenarios | | New anomalies | | Easiness | |
|---|---|---|---|---|---|
| Datasets | | MVTec | Magnetic | MVTec | Magnetic |
| Supervised | | 0.048 | 0.034 | 0.035 | 0.025 |
| Negative Supervised | | 0.074 | 0.025 | 0.049 | 0.034 |
| Negative OCC | | 0.034 | 0.025 | 0.028 | 0.026 |
| Unsupervised OCC | | 0.038 | 0.024 | 0.034 | 0.029 |
| FixMatch | | 0.033 | 0.025 | 0.037 | 0.034 |
| SPADE (Ours) | | 0.041 | 0.032 | 0.032 | 0.025 |

Table 10: Standard deviations of experiments on image domain with (left) new types of anomalies, (right) labeling based on easiness scenarios in terms of overall AUC.

| Datasets | | Thyroid | | | | Drug | | | | Covertype | | |
|---|---|---|---|---|---|---|---|---|---|---|---|---|
| Metrics (AUC) | | Overall | Given | Missed | | Overall | Given | Missed | | Overall | Given | Missed |
| Negative Supervised | | 0.028 | 0.001 | 0.040 | | 0.011 | 0.013 | 0.014 | | 0.001 | 0.000 | 0.002 |
| Negative OCC | | 0.007 | 0.018 | 0.003 | | 0.020 | 0.021 | 0.020 | | 0.001 | 0.001 | 0.001 |
| Unsupervised OCC | | 0.016 | 0.016 | 0.017 | | 0.016 | 0.016 | 0.020 | | 0.001 | 0.001 | 0.001 |
| Weighted Elkanoto (Elkan & Noto, 2008) | | 0.022 | 0.035 | 0.026 | | 0.018 | 0.022 | 0.021 | | 0.006 | 0.006 | 0.010 |
| BaggingPU (Mordelet & Vert, 2014) | | 0.029 | 0.019 | 0.036 | | 0.019 | 0.020 | 0.020 | | 0.021 | 0.016 | 0.027 |
| SPADE (Ours) | | 0.042 | 0.001 | 0.060 | | 0.008 | 0.008 | 0.016 | | 0.002 | 0.001 | 0.002 |

Table 11: Standard deviations of the experiments on PU settings on 3 tabular datasets in AUC of overall/given/missed (not given).

| Datasets | | Kaggle Credit Fraud | | Xente Fraud | |
|---|---|---|---|---|---|
| Labeling ratio | | 5% | 10% | 10% | 20% |
| Supervised | | 0.002 | 0.001 | 0.024 | 0.009 |
| Negative Supervised | | 0.002 | 0.002 | 0.022 | 0.012 |
| OCC | | 0.021 | 0.043 | 0.064 | 0.010 |
| Negative OCC | | 0.011 | 0.007 | 0.005 | 0.010 |
| Unsupervised OCC | | 0.004 | 0.004 | 0.090 | 0.011 |
| VIME | | 0.012 | 0.013 | 0.023 | 0.019 |
| DANN | | 0.033 | 0.027 | 0.013 | 0.021 |
| SPADE (Ours) | | 0.001 | 0.001 | 0.001 | 0.009 |

Table 12: Standard deviations of the experiments with two real-world fraud detection datasets in terms of overall AUC.

# D  Additional Experiments

## D.1  Labeling high-risk samples

In this subsection, we evaluate the performance of SPADE in PNU settings only with the labeled high-risk samples which is a common practical setting in fraud detection applications (including anti-money laundering). More specifically, the predictive model first estimates the anomaly scores of the unlabeled data. Then, the users manually check the samples only with high anomaly scores, and label them as either positive or negative. Thus, most labeled samples are actually high-risk samples and most unlabeled samples are low-risk samples which make the distribution differences between labeled and unlabeled data.

Similar with easiness scenario, we first train a simple logistic regression model (with the full label) and compute the anomaly scores of the unlabeled data. Then, we only extract the high risk samples (e.g., with top 2% highest anomaly scores). Then, we provide true labels for 50% (selected by uniformly random) of those high risk samples. It means that we have 1% of labeled data (either positive or negative) and 99% of unlabeled data. We exclude original OCC as the baseline because in some cases, there are too small numbers of negatively labeled samples which make OCC hard to converge.

| Datasets | Thyroid | | | Drug | | | Covertype | | |
|---|---|---|---|---|---|---|---|---|---|
| Labeling ratio | 1% | 1.5% | 2.5% | 1% | 1.5% | 2.5% | 1% | 1.5% | 2.5% |
| Supervised | 0.758 | **0.984** | **0.984** | 0.578 | 0.655 | 0.615 | 0.619 | 0.602 | 0.669 |
| Negative Supervised | 0.726 | 0.814 | 0.905 | 0.697 | 0.727 | 0.778 | 0.635 | 0.667 | 0.734 |
| Negative OCC | 0.466 | 0.468 | 0.469 | 0.725 | 0.729 | 0.734 | 0.829 | 0.836 | 0.848 |
| Unsupervised OCC | 0.502 | 0.526 | 0.519 | 0.763 | 0.766 | 0.769 | 0.846 | 0.851 | **0.865** |
| VIME | 0.677 | 0.703 | 0.717 | 0.669 | 0.681 | 0.690 | 0.841 | 0.843 | 0.847 |
| DANN | 0.735 | 0.744 | 0.749 | 0.724 | 0.747 | 0.761 | 0.749 | 0.762 | 0.769 |
| SPADE (Ours) | **0.924** | 0.983 | 0.981 | **0.828** | **0.835** | **0.838** | **0.871** | **0.867** | **0.865** |

Table 13: Experimental results with labeling only on high-risk samples in terms of overall AUC.

Table 13 shows that SPADE achieves superior or similar anomaly detection performance compared to the best alternative.

## D.2 Additional ablation studies

| Scenarios | New anomaly types | | Easiness | | |
|---|---|---|---|---|---|
| Variants | Drug | Covertype | Thyroid | Drug | Covertype |
| (i) No partial matching | 0.827 | 0.916 | 0.811 | 0.830 | 0.869 |
| (ii) No ensemble | 0.830 | 0.915 | 0.786 | 0.830 | 0.876 |
| (iii) $\beta = 0$ (No self-supervised) | 0.829 | 0.919 | 0.818 | 0.827 | 0.877 |
| (iv) No normal samples | 0.835 | 0.922 | 0.822 | 0.841 | 0.887 |
| (v) Majority vote | 0.835 | 0.918 | 0.807 | 0.839 | 0.890 |
| **SPADE** | 0.837 | 0.928 | 0.833 | 0.846 | 0.892 |

Table 14: Ablation studies on multiple tabular datasets with new anomaly and easiness settings: (i) without partial matching, (ii) without an ensemble of OCC, (iii) with $\beta = 0$ (No self-supervised learning), (iv) without normal samples for pseudo-labeler training, (v) majority vote instead of unanimous votes for pseudo-labeling.

### D.3   Additional sensitive analyses on $\beta$

In this subsection, we provided additional sensitive analyses on the important hyper-parameter ($\beta$) using three tabular datasets with new anomaly settings.

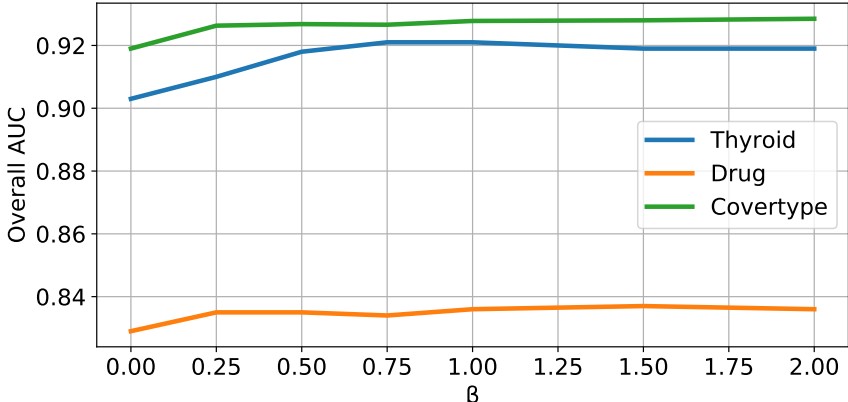

Figure 6: Overall AUC across different values of $\beta$ using three tabular datasets. ($\beta = 0$ represents SPADE without self-supervised learning.)

