# OpenReview forum: "SPADE: Semi-supervised Anomaly Detection under Distribution Mismatch"
_TMLR — Accepted by TMLR_

### Review · Reviewer_52Xa · 2022-11-30

**Summary Of Contributions:**

This paper presents work on semi-supervised anomaly detection, where labeled and unlabeled data are allowed to come from different distributions. A framework SPADE is proposed accordingly. SPADE shows state-of-the-art performance across multiple scenarios with distribution mismatch in various domains, e.g., the tabular and image.

**Audience:**

Yes

**Broader Impact Concerns:**

This paper did not provide discussions on broader impacts.

**Claims And Evidence:**

Yes

**Requested Changes:**

- Is this work the first to target the problem, i.e., semi-supervised anomaly detection under distribution mismatch? There are some discussions on related work. However, there is no discussion about this.
- The contributions can be reorganized. Some contributions are only the procedures of the proposed method.
- The paper claims that OCC-based pseudo-labeling is better suited when a distribution mismatch exists between labeled and unlabeled data. Is there some evidence to support this claim?
- The paper argues that, with a small amount of labeled data (less common in domain adaptation settings), the performance of the trained model on a small source data would be limited. For the proposed method of this paper, how to address the issue of insufficient labeled data at a high level? It is not clear. Besides, could the paper provide intuitions about why the proposed method outperforms previous works?
- To determine the parameters $\eta^p$ and $\eta^n$, the partial matching method is employed. It seems that the method only can help to identify easy outliers. How to address the problem? Additionally, how to determine $\alpha$ and $\beta$ in practice? Are there ablation studies for their sensitivity?
- In partial cases, some baselines outperform the proposed method of this paper. The baselines are not designed for the specific problem in this paper. Could the paper provide more explanations about why they can achieve better performance?

**Strengths And Weaknesses:**

**Strengths**

- The motivation is strong and clear. The research problem is realistic and important.
- Experiments are overall convincing. In partial cases, the performance improvement is clear.

**Weaknesses**

- The proposed method is complicated. There are multiple modules in the proposed method of this work. It needs to balance carefully different modules.
- Writing needs to be improved. Many key parts lack discussion and description. More details are provided below.

---

> ### Author Response · Authors · 2022-12-15
> **Response to the reviewer 52Xa's comments (1/2)**
>
> Thanks for all your valuable feedback that has helped us to improve our manuscript. Please see the detailed responses below and let us know if you have any further questions or comments.
>
> **Answer 1:** To the best of our knowledge, our paper is the first to tackle the general distribution mismatch scenarios for anomaly detection. There are some previous works that address specific scenarios of distribution mismatch such as PU or OOD settings (as described in related works), however, not the general cases. On the other hand, the distribution mismatch settings that are tackled in our paper cover all those specific settings.
>
> **Answer 2:** Thanks for noting this aspect. We have rewritten the contributions in the Introduction section to better emphasize the differentiated methodological aspects rather than the procedures.
>
> **Answer 3:** Thanks for raising this important aspect - clarifying this is of high value for our paper.
>
> As described with Fig. 2 and in Introduction, if there is distribution mismatch, heavily relying on the labeled data or training with noisy labeled data can be suboptimal (as can be seen in the decision boundaries of Fig. 2(b, c)). On the other hand, with OCCs (without using the labeled data at all), we can achieve quite reasonable decision boundaries (Fig. 2(d)) - still not perfect due to not using labeled information.
>
> In SPADE, we incorporate these motivations and construct the pseudo-labeler in a way that it relies less on the labeled data. More specifically, when constructing the OCCs, our framework excludes the positive labeled data to avoid overfitting to a small number of positive labeled data. In addition, we use the consensus approach on pseudo-labeling to significantly reduce the label noise in pseudo-labeled samples. As such, SPADE can identify new types of anomalies (see Table 2 - “Missed” columns) and can generalize better to when there is a distribution mismatch.
>
> **Answer 4:**  One of the arguments is about the impact of noisy labels in the training data that might come from not treating the positive/negative/unlabeled samples distinctly. This impact is especially crucial when the amount of labeled data would be limited.
>
> In SPADE, we employ pseudo-labelers to create additional labeled data in a way. Then, using both small labeled data and additionally created pseudo-labeled data, we train the predictive model that can perform better than alternatives. Thus, improvements in the predictive model rely on high quality pseudo-labels, provided by the proposed pseudo-labeler model.
>
> The fundamental intuitions on the outperformance of SPADE can be summarized as:
> - Supervised learning methods cannot extract the information from unlabeled data, that can be very large in proportion.
> - General semi-supervised learning methods are not designed for anomaly detection or distribution mismatch; they heavily rely on the labeled data which do not come from test distributions.
> - OCC-based methods cannot fully utilize the given labeled data, and treating negative and unlabeled data in the same way can be considered as label noise.
> - Proposed pseudo-labeler is very high performant, yielding very high accuracy for pseudo-labels compared to their ground truth.
>
> Please see more detailed comparisons in Table 1.
>
> **Answer 5:** Regarding how effective the proposed mechanism of SPADE is in identifying difficult outliers, we already have an analysis. Table 2 shows that, with SPADE, we can identify the outliers which are not given in the training data (see “Missed” columns), i.e. more “difficult” outliers.
>
> We agree that our partial matching method and consensus rule of the pseudo-labelers make more conservative decisions for additional pseudo-labeled data. But as can be seen in Table 7 and 14, with majority vote (less conservative decision of pseudo-labeling), the performances are much worse than consensus approach. This would be the strong support that more conservative pseudo-labeling (thus, less label noise in the pseudo-labeled samples) makes more robust and accurate semi-supervised anomaly detection models.
>
> As explained in Sec. 4.5 (just above the Algorithm 1) and Appendix B.5, we set $\alpha$ and $\beta$ to their default value ($1.0$) in all the experiments. For $\alpha$, we have ablation studies in Fig. 5. For $\beta$, we have included additional sensitive analyses in Appendix D.3 in the revised manuscript. The performances don’t seem to be sensitive to either $\alpha$ or $\beta$ and setting them as $1.0$ gives a decent result across all cases.

---

> > ### Author Response · Authors · 2022-12-15
> > **Response to the reviewer 52Xa's comments (2/2)**
> >
> > **Answer 6:** Note that for the vast majority (>90%) of the cases, our method SPADE outperforms alternatives. In addition, for more than 50% of them, the performance improvements are significant. For some cases (e.g., Table 2 - Covertype - Given), supervised model is only focused on making decision boundaries of the given labeled data; thus, it may work better for discovering given anomaly types. But in terms of overall performance, the supervised model is much worse than SPADE. In case of Table 4 - Drug, the performance difference is very small between the best model and SPADE. We attribute this to the distribution mismatch being small.
> >
> > One of the major motivations of our paper is to underline the importance of developing a tailored methodology for this important problem, rather than adopting the popularly-used methods in the literature. We focused on the baselines that correspond to popularly-used methods, and specifically-designed methods for the corresponding problem like PU learning.
> >
> > --------------------Answers to the two weakness points--------------------
> >
> > **Answer 7:** SPADE combines different important modules to push the state-of-the-art in semi-supervised anomaly detection. In Sec. 6 (Table 7) and Table 14, we present ablation studies to show the importance of each module, and simpler versions of SPADE without them would be worse in terms of anomaly detection performance.
> >
> > Although SPADE combines multiple modules, indeed it is a quite robust and easy-to-adapt method. As explained in Answer 5, its performance is not sensitive to hyperparameters other than $\eta^p$ and $\eta^n$, for which we propose a partial matching method to pick them without relying on a labeled dataset.
> >
> > **Answer 8:** Thank you for the suggestion. We have revised the introduction, especially on the contribution parts. Also, we have added more discussions about sensitivities of the hyper-parameters, details of OCC model training, and motivations of using SPADE for distribution mismatch scenarios in the revised manuscript.

---

### Review · Reviewer_LKzW · 2022-12-04

**Summary Of Contributions:**

This paper proposes a semi-supervised ensemble method for anomaly detection. In contrast to the previous methods that often assume that the labeled and unlabeled data come from the same distribution, this paper proposes a method that shows empirical robustness even under distribution mismatch.

**Audience:**

Yes

**Claims And Evidence:**

Yes

**Requested Changes:**

- The author should explain more on why the proposed method can address or alleviate the mismatch issues, and which part of the method is mainly designed to do so, according to the ablation study, each component boosts the performance, and it remains unclear which components can address mismatch, and if all of them are able to do so, please provide an explanation on why they are able to do so.

- In section 4.4, can you describe in detail how did you use Otsu's method to identify the threshold, and how did you calculate the Wasserstein distance, I feel there are many details that are skipped by the authors.

- I realized that in Algorithm 1, g, h, and q are initialized, is there a specific reason for doing this, wondering what would happen if they were continuously trained.

**Strengths And Weaknesses:**

**Strengths**:

- Overall, this paper is well-written, and easy to follow, even though I am not extremely familiar with the anomaly detection task.

- The idea is simple. Ensembling multiple one-class classifiers for anomaly detection improves the robustness of the method and sounds rather natural to me.  The final detector is an integration of the K detectors so that the final performance is improved.

- The effectiveness of the proposed method is validated on different datasets, from synthetic to real-world data, and across tabular to image datasets. Ablation studies also isolate the effects of each component of the final SPADE method.

**Weakness**:

- The major concern stems from the weak link between the distribution mismatch and the proposed method. The whole paper only states what the authors have done but does not mention why doing so can address the issues. Specifically, It remains very unclear to me why the distribution mismatch can be addressed by the proposed method, and which component of the method is the main block design for the distribution mismatch.

- Following the previous concern, it is widely known that consensus can improve robustness, and it is unclear whether this can be claimed as a novelty to the paper, also the author did not illustrate clearly why consensus achieves better robustness specifically for distribution mismatch.

---

> ### Author Response · Authors · 2022-12-15
> **Response to the reviewer LKzW's comments**
>
> Thanks for all your valuable feedback that has helped us to improve our manuscript. Please see the detailed responses below and let us know if you have any further questions or comments.
>
> **Answer 1:** Thanks for raising this important aspect - clarifying this is of high value for our paper.
>
> As described with Fig. 2 and in Introduction, if there is distribution mismatch, heavily relying on the labeled data or training with noisy labeled data can be suboptimal (as can be seen in the decision boundaries of Fig. 2(b, c)). On the other hand, with OCCs (without using the labeled data at all), we can achieve quite reasonable decision boundaries (Fig. 2(d)) - still not perfect due to not using labeled information.
>
> In SPADE, we incorporate these motivations and construct the pseudo-labeler in a way that it relies less on the labeled data. More specifically, when constructing the OCCs, our framework excludes the positive labeled data to avoid overfitting to a small number of positive labeled data. In addition, we use the consensus approach on pseudo-labeling to significantly reduce the label noise in pseudo-labeled samples. As such, SPADE can identify new types of anomalies (see Table 2 - “Missed” columns) and can generalize better to when there is a distribution mismatch.
>
> **Answer 2:** Indeed, using consensus is not the main contributor of the SPADE’s anomaly detection performance, the design of pseudo-labelers and how we utilize them for data refinement are more critical.
>
> As explained above, we significantly reduce the dependency of the labeled data when constructing the OCC models via the pseudo-labelers (by excluding the positive labeled data; unlabeled data samples are much larger than labeled negative samples when training the OCC models). In that case, we can construct more robust decision boundaries that can identify the pseudo-negative and pseudo-positive samples without overfitting to the labeled data, a critical aspect when there is distribution mismatch. Of course, the robustness of pseudo-labelers is also contributed by the fact that we employ an ensemble (with consensus approach) as noted, but it is not the main contribution for addressing the distribution mismatch.
>
> We have clarified these points in the revised manuscript - we hope these resolve your concern.
>
> **Answer 3:** Please see Answer 1 & 2.
>
> **Answer 4:** Otsu’s method is applied to one-dimensional anomaly scores. For all unlabeled samples, we extract one-dimensional anomaly scores from the trained OCCs. Then, we find the threshold that minimizes the intra-class variances of two subgroups (splitted by the threshold) of anomaly scores. More details can be found here: https://en.wikipedia.org/wiki/Otsu%27s_method. Note that in Otsu’s method, we do not use Wasserstein distance.
>
> Wasserstein distance is computed in Eqs. 3 and 4. Here, the outputs of $o_k$ are one-dimensional anomaly scores and we compute the Wasserstein distance between two one-dimensional anomaly scores. Wasserstein distances between two one-dimensional vectors can be computed as the integral of the cumulative distribution function differences. More details can be found here: https://en.wikipedia.org/wiki/Wasserstein_metric. We have added more details about Otsu’s method and Wasserstein distance computation in the revised manuscript.
>
> **Answer 5:** In Algorithm 1, Lines 1-9 correspond to the pseudo-code of the “pseudo-labeler function” and these lines are not executed unless the function is called in Line 13. So, the first executed line in Algorithm 1 is Line 10. We first initialize all the networks ($g, h, q$). Then, until $g, h, q$ are converged, we estimate the pseudo-labels of the unlabeled data (Line 13) and update $g, h, q$ based on the pseudo-labels (Line 14). Initialization happens only once at the beginning and $g, h, q$ are continuously trained afterwards until convergence. We have clarified Algorithm 1 to avoid possible misunderstandings in the revised manuscript.

---

> > ### Comment · Reviewer_LKzW · 2023-01-04
> > **Response**
> >
> > Thanks for the response. Most of my concerns have been solved.

---

### Review · Reviewer_Jqdn · 2022-12-05

**Summary Of Contributions:**

The paper considers the problem of semi-supervised anomaly detection and proposes a novel framework SPADE which is not limited by the assumption that labeled and unlabeled data come from the same distribution. They introduce a pseudo-labeling mechanism using an ensemble of OCCs and combining supervised and self-supervised learning. In addition, their framework involves an approach to pick hyper-parameters without a validation set. Experiments show that SPADE outperforms the alternatives in various scenarios.

**Audience:**

Yes

**Claims And Evidence:**

Yes

**Requested Changes:**

1. The reason that using partial matching to determine the thresholds $\eta^p$ and $\eta^n$ is free of extra validation datasets should be further explained.
2. How to train the OCCs with negative labeled data and unlabeled data should be more described. What loss functions or methods are employed to train the OCCs?

**Strengths And Weaknesses:**

#Strengths

1. This paper focuses on the problem that distribution mismatch commonly in semi-supervised learning with labeled and unlabeled data coming from different distributions. It has great practical significance.
2. This paper proposes a new approach to select the hyper-parameters of thresholds in the design of the pseudo-labeler.
3. SPADE shows a significant improvement compared with other methods in multiple settings that represent common real-world scenarios.
4. The methods can be flexibly applied with several self-supervised learning methods to improve performance.

#Weaknesses

1. This paper claim that using partial matching to determine the thresholds $\eta^p$ and $\eta^n$ is free of extra validation datasets and it can reduce the number of labeled samples in training data which is critical in semi-supervised settings. But the selection of other hyper-parameters(such as the trade-off parameters $\alpha,\beta$  and the selection of models in the training phase) still needs a validation dataset. Further explanation about the validation dataset should be made.
2. In subsection 4.3, "Each OCC is trained with the negative labeled data and one of K disjoint subsets of unlabeled data", how to train the OCCs with negative labeled data and unlabeled data should be described minutely. What loss functions or methods are employed to train the OCCs?

---

> ### Author Response · Authors · 2022-12-15
> **Response to the reviewer Jqdn's comments**
>
> Thanks for all your valuable feedback that has helped us to improve our manuscript. Please see the detailed responses below and let us know if you have any further questions or comments.
>
> **Answer 1:** This is indeed a very critical point, thanks for bringing it up. For semi-supervised learning where we have access to a very small amount of labeled data for model development, it is important to minimize the dependence on the labeled validation data.
>
> As explained in Sec. 4.5 (just above the Algorithm 1) and Appendix B.5, we set both $\alpha$ and $\beta$ to their default value of $1.0$ in all experiments. As shown in Fig. 5, the performance of SPADE is not sensitive with different $\alpha$ values (except $\alpha=0$, no pseudo-labeling). In the revised manuscript (see Appendix D.3), we have included additional sensitive analyses on $\beta$. As can be seen from the additional results, SPADE is also not sensitive with different $\beta$ values (except $\beta=0$, no self-supervised learning). Thus, a validation dataset to optimize $\alpha$ and $\beta$ is not needed, although existence of it might provide slight benefit in performance via better selection of hyper-parameters.
>
> On the other hand, the performance of SPADE is indeed sensitive to $\eta^p$ and $\eta^n$. Proper optimization of $\eta^p$ and $\eta^n$ is critical for achieving the SOTA performance. If we have a validation set, we can use it for optimizing $\eta^p$ and $\eta^n$. Nevertheless, even without a validation set, we can utilize the proposed partial matching approach (overviewed in Sec. 4.4), as shown in Eqs. 3 and 4, to determine the critical hyper parameters $\eta^p$ and $\eta^n$ (while having fixed other hyper-parameters as described in Appendix B.5).
>
> **Answer 2:** To train OCCs, we only need data from a single class - we do not need label information. For pseudo-labeler of SPADE, we treat the negative labeled data and one of $K$ disjoint subsets of unlabeled data as the one-class data, to train the OCCs. As the OCC model, as explained in Appendix B.5, we use Gaussian Distribution Estimator (GDE) which utilizes one-class training data (negative labeled data + subsets of unlabeled data) to estimate the density function with maximum likelihood objective for the distribution assumption as the loss function. At inference, the likelihood outputs of GDE for each sample are used as the anomaly scores. We have added these details on OCC training in the revised manuscript (see Appendix B.5).

---

> > ### Comment · Reviewer_Jqdn · 2022-12-20
> > **Response**
> >
> > Thanks for the response. Some concerns have been solved.  There are some flaws in this paper:
> >
> > 1. The hyper-parameters associated with the method have been explained clearly and are indeed free of validation datasets. How about other hyper-parameters? Such as the learning rate and weight decay of a model, the selection of the training optimizer, or the model in which epoch during the training phase is selected as the final model. A validation dataset is still indispensable for these hyper-parameters unless the same hyper-parameters are used for all datasets.
> >
> > 2. This paper focuses on the problem that distribution mismatch commonly occurs in semi-supervised learning with labeled and unlabeled data from different distributions. It has great practical significance. However, there is no explanation for why the method is suitable for the distribution mismatch problem.

---

> > > ### Author Response · Authors · 2022-12-25
> > > **Thanks for your response to our rebuttal**
> > >
> > > Thanks for checking our responses. It’s great to hear that the previous responses resolved some of your concerns. Hopefully, the below responses will resolve your remaining concerns related to our paper.
> > >
> > > **Answer 1:** We appreciate that you find the explanations for selecting SPADE hyper-parameters (including $\alpha, \beta, \eta^p, \eta^n$) as it is a critical aspect for semi-supervised learning.
> > >
> > > Regarding the rest of the experiment parameters, we use the default values as follows.
> > > We use Adam optimizer with default learning rate (0.001).
> > > We set 20 epochs for training and apply early stopping based on the training loss (please see Appendix B.5 about the details of convergence criteria).
> > > We do not use any additional regularization like weight decay.
> > >
> > > We agree that the existence of the labeled validation set might provide a slight benefit in performance via better selection of various hyper-parameters. However, we also think that having the capability of training the model without a validation set would be important in some cases where having a labeled validation set may significantly decrease the labeled training set (e.g., with limited labeled data). In most real-world applications, the small amount of labeled data for model development would need to be shared between training, validation and test datasets, so having a method that has high labeled data requirements for validation would not be very practical.
> > >
> > > We have toned down our claim on the free of validation dataset in the revised manuscript and we will describe the point more explicitly as above.
> > >
> > > **Answer 2:** Thanks for recognizing the practical significance of our problem setting (semi-supervised settings with distribution mismatch).
> > >
> > > Indeed, the method’s suitableness for the distribution mismatch problem is highly important to clarify and we hope the below convey the key points.
> > >
> > > Primarily, as described with Fig. 2 and in Introduction, if there is distribution mismatch, heavily relying on the labeled data or training with noisy labeled data can be suboptimal (as shown in the decision boundaries of Fig. 2(b, c)). On the other hand, with OCCs (without using the labeled data at all), we can achieve quite reasonable and robust decision boundaries (Fig. 2(d)), although still not perfect due to not using labeled information. In SPADE, we incorporate these motivations and construct the pseudo-labeler in a way that it relies less on the labeled data. More specifically, when constructing the OCCs, SPADE excludes the positive labeled data to avoid overfitting to a small number of positive labeled data. In addition, SPADE utilizes unlabeled data samples much larger than labeled negative samples, when training the OCC models. In that case, we can construct more robust decision boundaries that can identify the pseudo-negative and pseudo-positive samples without overfitting to the labeled data, a critical aspect when there is distribution mismatch. We also use the consensus approach on pseudo-labeling to significantly reduce label noise in pseudo-labeled samples. As such, SPADE can identify new types of anomalies (see Table 2 - “Missed” columns) and can generalize better to when there is a distribution mismatch.
> > >
> > > We have clarified these points in the revised manuscript - we hope these resolve your concerns.

---

> > > > ### Comment · Reviewer_Jqdn · 2022-12-29
> > > > **Response**
> > > >
> > > > Thanks for the response. My concerns have been solved.

---

> > ### Public Comment · ~Claude_Ross1 · 2023-12-04
> > **Response**
> >
> > Once again, I appreciate your thoughtful and useful feedback. Each reviewer has recently received a copy of the updated manuscript along with our thorough answers. If the reviewers or editor have any further questions or comments https://basketballstarsgame.io , please let us know when you have seen the updated article and the comprehensive reply.

---

### Comment · Action_Editors · 2022-12-05
**Rolling discussions**

Dear authors,

Three reviews have been submitted. Can you provide rebuttals and discuss them with the reviewers in 2 weeks? The reviewers will submit formal recommendations after 2 weeks.

Best wishes,
Tongliang

---

> ### Author Response · Authors · 2022-12-06
> **Thank you for the insightful reviews**
>
> We will provide detailed rebuttals and the revised manuscript within 2 weeks.
> Thank you!

---

> > ### Author Response · Authors · 2022-12-15
> > **Submitted the revised manuscript and detailed responses to each reviewer**
> >
> > Thank you again for the insightful and helpful comments.
> > We just submitted the revised manuscript and detailed responses to each reviewer.
> > Please see the detailed responses and revised manuscript and let us know if reviewers or editor have any further questions or comments.

---

### Decision · Action_Editors · 2023-01-19

**Recommendation:** Accept as is

**Comment:**

This paper presents work on semi-supervised anomaly detection, where a semi-supervised ensemble method is proposed. The proposed method successfully avoids the assumption that the labeled and unlabeled data come from the same distribution. It has excellent potential to be applied in realistic scenarios. Extensive experiments also confirm the effectiveness of the method.

Before responses from authors, three reviewers pointed out several concerns including (1) the link between distribution mismatch and the proposed method; (2) the settings of hyper-parameters; (3) unclear descriptions. The authors provide detailed responses to address the concerns. All reviewers are satisfied with the responses and appreciate the contributions of this paper. We, therefore, recommend an acceptance.

This paper is high quality and then selected for the Featured Certification, which is recommended by most reviewers. After the responses from the authors, the claims of this paper are clearly justified. AE and reviewers consider that this paper makes sufficient contributions to the research area of semi-supervised anomaly detection, mainly including its strong motivation, practical problem setting, proposed novel framework, convincing experiments, and detailed analysis/discussions of experimental results. Therefore, it deserves the Featured Certification and should be highlighted.

**Audience:**

This paper will be of interest to researchers from the communities of semi-supervised learning and anomaly detection.

**Claims And Evidence:**

The writing and organization of this paper are satisfactory overall. Most claims are accurate, convincing, and with evidence.

---

> ### Author Response · Authors · 2023-02-04
> **Thank you all for the great and insightful comments for this paper.**
>
> Those comments were very valuable and helped improve the manuscript significantly.
> Also, we really appreciate that the editor and reviewers recognized this paper as the featured certificate.
>
> We just uploaded the final camera ready version of the paper.
> Please let us know if there is anything else that we need to do for finalizing this process.
>
> Thank you again.